# Bayesian Networks Framework for Estimating Individual Survival Distributions: An Application in Amyotrophic Lateral Sclerosis Disease

## Abstract

Accurate survival prediction is essential in many fields, particularly in healthcare, where estimating not only time-to-event but the entire individual survival distribution (ISD) can inform personalized decision-making. While existing models such as Cox proportional hazards, MTLR, and deep neural networks provide partial solutions, they often rely on strong parametric assumptions or lack interpretability. In this work, we propose a Bayesian network (BN)-based framework for survival prediction that explicitly models the joint distribution over covariates, survival time, and censoring status. We learn the structure of the BN using Gibbs sampling and parameterize the conditional distributions using flexible parametric models tailored to variable types. Producing the ISD for an instance is performed by inference, which is done via sampling over the Markov blanket of survival and censoring nodes, enabling efficient estimation of personalized survival curves. We evaluate our method on a real-world (Amyotrophic Lateral Sclerosis) dataset and compare against state-of-the-art baselines using metrics such as C-index, MAE, and D-calibration. Results show that our BN-based model offers competitive or superior performance in estimating full ISDs, while also providing interpretability.

## 1 Introduction

Survival analysis models the time until the occurrence of an event of interest (such as death), often dealing with censoring – where the learner knows only a lower bound on the time to the event. Some models estimate not only the expected survival time of each instance, but also the full distribution of possible survival times for that individual, known as the individual survival distribution (ISD) Haider et al. (2020) – see Fig. 2. This distribution allows for probabilistic reasoning about personalized outcomes, uncertainty quantification, and robust decision-making – which is useful for many clinical, financial, and industrial applications. Traditional models, such as Cox proportional hazards (CoxPH) Cox (1972) or Accelerated Failure Time (AFT) Kleinbaum & Klein (2012), make simplifying assumptions that may not hold in real-world, high-dimensional data, and often do not provide full ISD estimates. Bayesian networks (BNs) Koller & Friedman (2009) provide a powerful probabilistic framework to represent joint distributions over structured domains with conditional dependencies. Their graphical structure naturally captures complex inter-variable relationships and context-specific independencies. By incorporating survival time as a node within a BN, along with covariates and censoring indicators, the model can encode rich dependencies and support personalized inference. This structure enables the estimation of individual survival distributions via probabilistic queries conditioned on the local structure, such as the Markov blanket of the survival node.

Several studies have leveraged probabilistic graphical models for survival prediction. Early work includes dynamic Bayesian networks Jiang et al. (2014), piecewise exponential models Nagpal et al. (2021), and more recent efforts to embed BNs into clinical time-to-event models Kuan & Greiner (2021), Sheidaei et al. (2022), and Shahmirzalou et al. (2023). While deep learning-based survival models such as DeepSurv Katzman et al. (2018) and MTLR Yu et al. (2011) have shown promise in estimating ISDs, they often require large amounts of labeled data and lack interpretability. Bayesian

networks offer a compelling alternative by combining expressive modeling with interpretability and flexibility in incorporating prior knowledge. Kraisangka & Druzdzel (2014; 2018) proposed combining the CoxPH model with specific discrete Bayesian network structures, such as naive Bayes, without performing structure learning. However, the resulting models achieved inferior performance compared to the standard CoxPH method.

In this paper, we propose a novel BN-based approach to survival prediction that directly models the joint distribution over covariates, survival time, and censoring status. Four types of conditional probability distributions (CPDs) are considered for binary, (non-binary) categorical, survival time and non-survival-time continuous variables. We parameterize conditional distributions based on variable types, including exponential models (motivated by CoxPH, Deepsurv, and AFT) for survival time, and learn the BN structure using Gibbs sampling. Then the parameters are learned by maximum likelihood estimation (MLE). Finally, for producing the survival curve, inference is performed by sampling over the Markov blanket of the survival and censoring indicator nodes, allowing estimation of the full individual survival distribution. We evaluate our method on real-world survival datasets and compare against state-of-the-art models using metrics such as C-index, mean absolute error, and D-calibration Haider et al. (2020). Our results demonstrate that the BN-based model offers competitive performance while retaining interpretability and distributional insight.

## 2 PRELIMINARIES

This section introduces the foundational concepts of survival analysis, Bayesian networks, and the specific modeling techniques that form the basis of our work.

### 2.1 THE FORMALISM OF SURVIVAL ANALYSIS

In survival analysis, we model the time until a specific event occurs. For each individual $i$ in a population, this is defined by two key random variables.

**Definition 1 (Event and Censoring Times)** *For each individual $i$, the* event time $E_i$ *is the time at which the event of interest (*e.g.*, death, system failure) occurs. The* censoring time $C_i$ *is the time beyond which the individual is no longer observed or followed up, so whether or when the event occurs after $C_i$ is unknown.*

In practice, we only observe the minimum of these two times, denoted as $T_i = \min(E_i, C_i)$. To distinguish between observed events and censored observations, we use an event indicator $\delta_i = \mathbb{I}(E_i \leq C_i)$, where $\delta_i = 1$ signifies that the event was observed at time $E_i$, and $\delta_i = 0$ signifies that the observation was right-censored at time $C_i$. The primary objective is to estimate the probability of an individual surviving past a certain time $t$.

**Definition 2 (Individual Survival Distribution)** *The **survival function**, $S(t)$, is the probability that an individual's event time $E$ is greater than some time $t$:*

$$S(t) = P(E > t).$$

*For personalized predictions, this function is conditioned on a vector of individual-specific covariates $\mathbf{X}_i$. This yields the **Individual Survival Distribution (ISD)**, denoted $S_i(t)$:*

$$S_i(t) = P(E_i > t \mid \mathbf{X}_i).$$

*which models the unique survival probability trajectory for individual $i$.*

A critical assumption underpinning most survival models is that of **conditional independent censoring**. This assumption posits that, given an individual's covariates $\mathbf{X}_i$, their true event time $E_i$ is statistically independent of their censoring time $C_i$. Formally, this is expressed as:

$$E_i \perp C_i \mid \mathbf{X}_i.$$

This assumption is crucial because it allows the likelihood of the observed data to be formulated without needing to explicitly model the censoring mechanism itself. Under this assumption, we can proceed to define the likelihood function for the model parameters.

**The Survival Likelihood Function.** Given a dataset of $n$ individuals $(\mathbf{t}, \boldsymbol{\delta}, \mathbf{X})$ and model parameters $\boldsymbol{\theta}$, the likelihood is given by the product of the probability densities for uncensored individuals and the survival probabilities for censored individuals:

$$L(\boldsymbol{\theta}; \mathbf{t}, \boldsymbol{\delta}, \mathbf{X}) \;=\; \prod_{i=1}^{n} [f(t_i \mid \mathbf{X}_i; \boldsymbol{\theta})]^{\delta_i} [S(t_i \mid \mathbf{X}_i; \boldsymbol{\theta})]^{1-\delta_i}, \tag{1}$$

where $t_i$ represents the observed time $T_i$, and $f(t \mid \mathbf{X}_i; \boldsymbol{\theta}) = -\frac{d}{dt} S(t \mid \mathbf{X}_i; \boldsymbol{\theta})$ is the conditional probability density function. This formulation ensures that censored observations contribute information by constraining the survival function to be above a certain value at the time of censoring.

## 2.2 PROBABILISTIC MODELING WITH BAYESIAN NETWORKS

**Definition 3 (Bayesian Network)** *A Bayesian Network (BN) is a probabilistic graphical model that represents a joint probability distribution over a set of random variables $\mathcal{X} = \{X_1, \ldots, X_d\}$. It is a Directed Acyclic Graph (DAG) $\mathcal{G} = (\mathcal{V}, \mathcal{E})$, where each node in $\mathcal{V}$ correspond to a variable in $\mathcal{X}$ and directed edges $\mathcal{E}$ represent conditional dependencies of these variables.*

The structure of the BN encodes conditional independence assumptions, allowing the joint probability distribution to be factorized as a product of local CPDs:

$$P(\mathcal{X}) \;=\; \prod_{j=1}^{d} P(X_j \mid \mathrm{Pa}(X_j)),$$

where $\mathrm{Pa}(X_j)$ is the set of parent nodes of $X_j$ in $\mathcal{G}$. A key concept for efficient inference in Bayesian networks is the Markov blanket. Formally, it is defined by the local structure surrounding a node.

**Definition 4 (Markov Blanket)** *For any node $X_j$ in a Bayesian network $\mathcal{G}$, its **Markov Blanket**, denoted $\mathrm{MB}(X_j)$, is the following set of nodes:*

- *The parents of $X_j$,*

- *The children of $X_j$,*

- *The other parents of $X_j$'s children (aka "co-parents").*

This set has the crucial property that it is the minimal set of nodes that renders $X_j$ conditionally independent of all other nodes in the network. This property implies that:

$$P(X_j \mid \mathcal{X} \setminus \{X_j\}) \;=\; P(X_j \mid \mathrm{MB}(X_j)).$$

This allows for local computation, as predicting the posterior distribution of a variable only requires knowledge of the values of its Markov blanket. In our context, this means the survival distribution for an individual can be inferred by conditioning only on the variables in the Markov blanket of the event time node $E$.

## 2.3 MODEL SPECIFICATION AND LEARNING

**Structure Learning.** The DAG structure $\mathcal{G}$ is typically unknown and must be learned from data $\mathcal{D}$. We adopt a Bayesian approach where we aim to find the posterior distribution over graph structures:

$$P(\mathcal{G} \mid \mathcal{D}) \;\propto\; P(\mathcal{D} \mid \mathcal{G}) \, P(\mathcal{G}))$$

Prior distribution usually consider a uniform distribution $(P(\mathcal{G}) = Cte)$, and the marginal likelihood $P(\mathcal{D} \mid \mathcal{G})$ is often approximated using a score function. We use the Bayesian Information Criterion (BIC), which balances model fit and complexity:

$$\mathrm{BIC}(\mathcal{G}) \;=\; \log P(\mathcal{D} \mid \hat{\boldsymbol{\theta}}_{\mathcal{G}}, \mathcal{G}) \;-\; \frac{|\boldsymbol{\theta}_{\mathcal{G}}|}{2} \log N,$$

where $\hat{\boldsymbol{\theta}}_{\mathcal{G}}$ is the maximum likelihood estimate of the parameters for graph $\mathcal{G}$, $|\boldsymbol{\theta}_{\mathcal{G}}|$ is the number of parameters, and $N$ is the sample size. We employ Gibbs sampling to explore the space of DAGs and approximate the posterior distribution, where each step involves proposing local changes to the graph (*e.g.*, adding, removing, or reversing an edge) and accepting them based on the change in BIC score, while ensuring acyclicity.

## 2.4 PARAMETRIC AND SEMI-PARAMETRIC SURVIVAL MODELS

We now describe the specific survival models used as CPDs within our framework.

### 1. COX PROPORTIONAL HAZARDS (COXPH) MODEL

The CoxPH model Cox (1972) is a semi-parametric model that focuses on the hazard function, which represents the instantaneous rate of event occurrence at time $t$.

**Definition 5 (Cox Proportional Hazards Model)** *The hazard function $h(t \mid \mathbf{x})$ for an individual with covariates $\mathbf{x}$ is modeled as:*

$$h(t \mid \mathbf{x}) = h_0(t) \cdot \exp(\boldsymbol{\beta}^\top \mathbf{x}),$$

*where $h_0(t)$ is a non-parametric baseline hazard function and $\boldsymbol{\beta}$ is a vector of regression coefficients. The corresponding survival function is $S(t \mid \mathbf{x}) = \exp\left(-H_0(t) \cdot \exp(\boldsymbol{\beta}^\top \mathbf{x})\right)$, where $H_0(t) = \int_0^t h_0(s)\,ds$ is the cumulative baseline hazard.*

**Assumption 1 (Proportional Hazards)** *The core assumption of the CoxPH model is that the ratio of hazards for any two individuals is constant over time. This implies that covariates have a multiplicative and time-invariant effect on the hazard.*

**Parameter Estimation.** The coefficients $\boldsymbol{\beta}$ are estimated by maximizing the partial likelihood, which avoids specifying the baseline hazard $h_0(t)$. For a dataset where $T_i$ are the ordered event times, the log-partial likelihood is:

$$\ell(\boldsymbol{\beta}) = \sum_{i:\delta_i=1} \left[ \boldsymbol{\beta}^\top \mathbf{x}_i - \log \left( \sum_{j \in R(T_i)} \exp(\boldsymbol{\beta}^\top \mathbf{x}_j) \right) \right], \tag{2}$$

where $R(E_i) = \{k \mid T_k \geq E_i\}$ is the set of individuals "at risk" of an event at time $E_i$.

**Baseline Hazard Estimation.** After estimating $\hat{\boldsymbol{\beta}}$, the baseline cumulative hazard $H_0(t)$ is non-parametrically estimated using the Breslow estimator Breslow (1974):

$$\hat{H}_0(t) = \sum_{T_i \leq t} \frac{\delta_i}{\sum_{j \in R(T_i)} \exp(\hat{\boldsymbol{\beta}}^\top \mathbf{x}_j)}. \tag{3}$$

### 2. DEEPSURV

DeepSurv extends the CoxPH model to capture non-linear covariate effects. It replaces the linear term $\boldsymbol{\beta}^\top \mathbf{x}$ with a deep neural network $f_\theta(\mathbf{x})$, parameterized by $\theta$.

**Definition 6 (DeepSurv)** *The hazard function is given by:*

$$h(t \mid \mathbf{x}) = h_0(t) \cdot \exp(f_\theta(\mathbf{x})).$$

DeepSurv retains the proportional hazards assumption but allows the risk score $f_\theta(\mathbf{x})$ to be a highly non-linear function of the covariates. The model is trained by maximizing the same log-partial likelihood as in CoxPH, with the network's output replacing the linear predictor:

$$\ell(\theta) = \sum_{i:\delta_i=1} \left[ f_\theta(\mathbf{x}_i) - \log \left( \sum_{j \in R(T_i)} \exp(f_\theta(\mathbf{x}_j)) \right) \right]. \tag{4}$$

This formulation provides significantly greater flexibility in modeling complex dependencies between covariates and survival outcomes.

3. ACCELERATED FAILURE TIME (AFT) MODELS

AFT models offer an alternative perspective by modeling the event time directly.

**Definition 7 (Accelerated Failure Time Model)** *AFT models assume a linear relationship between the logarithm of the survival time and the covariates:*

$$\log(T) = \boldsymbol{\beta}^\top \mathbf{x} + \varepsilon,$$

*where $\varepsilon$ is a random error term from a specified probability distribution (*e.g., normal, logistic).

**Assumption 2 (Multiplicative Time Effect)** *The core assumption is that covariates act multiplicatively on the time scale, thus "accelerating" or "decelerating" the time to event. For example, a positive coefficient accelerates the failure process (shortens survival time).*

Unlike CoxPH, AFT models are fully parametric. If $\varepsilon$ follows a standard normal distribution $\mathcal{N}(0, \sigma^2)$, the survival function for the corresponding log-normal AFT model is:

$$S(t \mid \mathbf{x}) = 1 - \Phi\left(\frac{\log(t) - \boldsymbol{\beta}^\top \mathbf{x}}{\sigma}\right),$$

where $\Phi(\cdot)$ is the standard normal CDF. Parameters $(\boldsymbol{\beta}, \sigma)$ are estimated via maximum likelihood using the full survival likelihood, providing an interpretable framework where covariate effects relate directly to survival time.

# 3 PROBLEM FORMULATION AND METHODOLOGY

## 3.1 SURVIVAL PREDICTION VIA BAYESIAN NETWORKS

We address the problem of personalized survival prediction by learning a probabilistic graphical model from data. Given a dataset $\mathcal{D} = \{(\mathbf{x}_i, t_i, \delta_i)\}_{i=1}^{N}$ of $N$ individuals, where $\mathbf{x}_i \in \mathbb{R}^d$ is a vector of covariates, $t_i = \min(E_i, C_i)$ is the observed time, and $\delta_i \in \{0, 1\}$ is the event indicator, our goal is to learn a Bayesian Network that models the underlying data-generating process.

Formally, we seek to learn a BN $\mathcal{B} = (\mathcal{G}, \boldsymbol{\theta})$, where $\mathcal{G}$ is a DAG over the random variables $\mathcal{X} = \{X_1, \ldots, X_d, E, \delta\}$, and $\boldsymbol{\theta}$ represents the parameters of the CPDs associated with each node in $\mathcal{G}$. The learned model will be used to estimate the ISD, $S(t \mid \mathbf{x}) = P(E > t \mid \mathbf{x})$, for a new individual with covariates $\mathbf{x}$.

## 3.2 HYBRID MODEL: NODE-SPECIFIC CPDS

To accommodate the heterogeneous nature of clinical or real-world data, we define a hybrid BN where the functional form of each CPD, $P(\text{Node} \mid \text{Pa}(\text{Node}))$, is selected based on the variable type. The parent set $\text{Pa}(\cdot)$ for each node is determined by the learned graph structure $\mathcal{G}$.

- **Survival Time (E):** The distribution of the event time $E$ is conditioned on its parents $\text{Pa}(E)$. We model its survival function $S(t \mid \text{Pa}(E))$ using one of three established survival models, allowing for flexible dependency modeling:
  - *Cox Proportional Hazards (CoxPH):*

  $$P(t \mid \text{Pa}(E)) = \exp\left(-\hat{H}_0(t) \cdot \exp(\boldsymbol{\theta}^\top \text{Pa}(E))\right)$$

  where $\hat{H}_0(t)$ is the Breslow estimate of the baseline cumulative hazard (see Eq. 3) learned from the training data.
  - *DeepSurv:* A non-linear extension of the CoxPH model.

  $$P(t \mid \text{Pa}(E)) = \exp\left(-\hat{H}_0(t) \cdot \exp(f_\theta(\text{Pa}(E)))\right)$$

  where $f_\theta$ is a deep neural network parameterized by $\theta$.

---

**Algorithm 1** Structure Learning via Metropolis-Hastings

---

**Input**: Dataset $\mathcal{D}$, Max iterations $T_{\max}$.
**Output**: Highest-scoring DAG $\mathcal{G}^\star$.

1: Initialize $\mathcal{G}^{(0)}$ as a random DAG; set $\mathcal{G}^\star \leftarrow \mathcal{G}^{(0)}$.
2: **for** $t = 1$ **to** $T_{\max}$ **do**
3:    Propose a new graph $\mathcal{G}_{prop}$ from the neighborhood $\mathcal{N}(\mathcal{G}^{(t)})$ by applying a single valid edge operation (add, delete, or reverse).
4:    Calculate the local change in score $\Delta\text{BIC} = \text{BIC}(\mathcal{G}_{prop}) - \text{BIC}(\mathcal{G}^{(t)})$.
5:    Accept $\mathcal{G}_{prop}$ to form $\mathcal{G}^{(t+1)}$ with probability $\alpha = \min(1, \exp(\Delta\text{BIC}))$.
6:    If accepted, $\mathcal{G}^* \leftarrow \mathcal{G}^{(t)}$
7:    If not accepted, set $\mathcal{G}^{(t+1)} \leftarrow \mathcal{G}^{(t)}$.
8: **end for**
9: **return** $\mathcal{G}^\star$

---

   – *Accelerated Failure Time (AFT):* A log-linear model on time.

$$P(t \mid \text{Pa}(E)) = 1 - \Phi\left(\frac{\log(t) - \boldsymbol{\beta}^\top \text{Pa}(E)}{\sigma}\right)$$

   where $\Phi$ is the standard normal CDF, corresponding to a log-normal AFT model.

- **Continuous Variables** ($X_j$): Modeled using a linear Gaussian CPD, where the mean is a linear function of its parents:

$$P(X_j \mid \text{Pa}(X_j)) = \mathcal{N}\left(X_j \mid \boldsymbol{\theta}_j^\top \text{Pa}(X_j), \sigma_j^2\right).$$

- **Binary Variables** ($X_j$): Modeled using logistic regression:

$$P(X_j = 1 \mid \text{Pa}(X_j)) = \frac{1}{1 + \exp(-\boldsymbol{\theta}_j^\top \text{Pa}(X_j))}.$$

- **Non-binary Categorical Variables** ($X_j$): For variables with $K$ categories ($3 \leq K \leq 6$), we use multinomial logistic regression:

$$P(X_j = k \mid \text{Pa}(X_j)) = \frac{\exp(\boldsymbol{\theta}_{jk}^\top \text{Pa}(X_j))}{\sum_{l=1}^{K} \exp(\boldsymbol{\theta}_{jl}^\top \text{Pa}(X_j))}.$$

### 3.3 STRUCTURE LEARNING VIA METROPOLIS-HASTINGS

Learning the optimal DAG $\mathcal{G}$ is an NP-hard problem due to the super-exponential size of the search space. We employ a Metropolis-Hastings (M-H) Hastings (1970) algorithm that uses a single-edge Gibbs-like proposal mechanism to explore the space of DAGs and identify a structure that maximizes the BIC score. The procedure is summarized in Algorithm 1.

This M-H sampler efficiently navigates the search space by leveraging local score updates. The acceptance rule ensures that the Markov chain converges to a stationary distribution concentrated on high-scoring structures. The highest-scoring DAG encountered, $\mathcal{G}^\star$, is returned for downstream inference.

**Remark 1** *A potential point of confusion is that our Bayesian Network is defined over the set of observed variables, including the observed time $t_i$ and the event indicator $\delta_i$, whereas the scientific goal is to estimate the distribution of the true, potentially unobserved, latent event time $T$. This raises the question of how querying the node for the observed time can yield the desired survival distribution for the true event time. The key to this approach lies in the parameter estimation for the time node's CPD, $P(E \mid \text{Pa}(E))$. The parameters of this CPD, $\boldsymbol{\theta}_E$, are not trained to simply model the raw distribution of the observed times $t_i$. Instead, they are optimized by maximizing the standard likelihood for right-censored survival data in (1). This objective function explicitly uses the event indicator $\delta_i$ to distinguish between observed events (fitting the probability density $f$) and*

*censored observations (fitting the survival function $S$). This training process ensures that the learned parameters $\hat{\boldsymbol{\theta}}_E$ are optimized to characterize the distribution of the true latent event time $TE$. The logic for handling censoring is thereby encoded directly into the time node's CPD. Consequently, when we perform inference on a new subject, querying the network for the distribution of the node corresponding to time correctly yields an estimate of the Individual Survival Distribution for the true event time, $P(E > t \mid \mathbf{x})$.*

### 3.4 INDIVIDUAL SURVIVAL DISTRIBUTION ESTIMATION

After learning the BN structure $\mathcal{G}^\star$ and its parameters $\hat{\boldsymbol{\theta}}$, we can then estimate the ISD for a new test subject with covariates $\mathbf{x}$. One issue is that the event indicator $\delta$ is unknown at prediction time. This suggests two principled methods for estimating the ISD.

**Method 1: Joint Inference over the Survival State.**   This is the most general approach, treating both the event time $E$ and the event indicator $\delta$ as unobserved random variables. To account for the full uncertainty, inference must be conditioned on the union of the Markov blankets of both nodes, excluding themselves. The predictive distribution is thus:

$$P(E \mid \mathbf{x}) = P(E \mid \mathbf{x}_{\mathrm{MB}(E) \cup \mathrm{MB}(\delta) \setminus \{E, \delta\}})$$

where $\mathbf{x}_{\mathrm{MB}(E) \cup \mathrm{MB}(\delta) \setminus \{E, \delta\}}$ denotes the observed covariate values. Due to the hybrid nature of our model, this distribution is approximated using Gibbs sampling, as detailed in Algorithm 2. This method provides a comprehensive prediction by marginalizing over the unknown event status.

**Method 2: Conditional Inference Assuming Event Occurrence.**   This is an efficient and highly practical alternative that directly estimates the survival distribution conditioned on the constraint that the event of interest has occurred (i.e., setting evidence $\delta = 1$). This approach aligns perfectly with our training procedure, where the parameters $\hat{\boldsymbol{\theta}}_T$ for the time node's CPD, $P(E \mid \mathrm{Pa}(E))$, are optimized to model the true (possibly latent) event time.

By fixing $\delta = 1$, we ask the model a precise and clinically relevant question: "For an individual with these characteristics, what is their survival trajectory given they have experienced the event?" Under this condition, inference simplifies to computing $P(E \mid \mathbf{x}, \delta = 1)$. By the properties of Bayesian networks, this is equivalent to conditioning on the variables within the Markov Blanket of the time node, $\mathrm{MB}(E)$, along with the evidence $\delta = 1$:

$$P(E \mid \mathbf{x}, \delta = 1) = P(E \mid \mathbf{x}_{\mathrm{MB}(E)}, \delta = 1)$$

We use a targeted sampling approach, as outlined in Algorithm 3, which is still more direct than the joint sampling of Method 1.

## 4   ALS DATASET AND EVALUATION

In this study, we applied our proposed method to estimate the ISD for each instance. The CALSNIC dataset which includes 213 ALS patients, with approximately 30 percent of instances censored. For each patient, it includes from 1 to 3 clinical visits, with 130 clinical features per visit. We consider two approaches to deal with multiple visits per patient: The first approach is to include only one visit per patient, which results in a smaller dataset. The second approach treats each visit as a separate instance, which may violate the assumption of independence between instances all visits of each patient are in the same fold. We also used two feature selection process to reduce the number of features. One based on domain knowledge (Table 3) and the other using dependency tests. To select the most relevant features, we performed independence tests between each feature and the survival time variable, using mutual information with a significance level. Therefore, there are four cases of the set of variables based on the number of visits that were considered and the types of feature selection.

We considered three different Bayesian Networks based on three types of CPD for the survival time variable, which are named BN-CoxPH, BN-DeepNN, and BN-AFT, and the standard survival prediction methods CoxPH, MTLR, DeepSurv, and AFT were evaluated on numerical ALS instances. We evaluated each model using the Concordance Index (C-index), MAE on Pseudo-Observations

Table 1: Survival prediction performance on ALS dataset (mean ± std across folds). Models with * used Method 2 for inference. Feature selection was based on domain knowledge.

| Method | MAE-PO ↓ | C-index ↑ | D-Cal. ↑ |
|---|---|---|---|
| **One Visit** | | | |
| BN-CoxPH | $0.23 \pm 0.03$ | $0.70 \pm 0.07$ | 1/5 |
| BN-CoxPH* | $0.23 \pm 0.04$ | $0.71 \pm 0.05$ | 2/5 |
| BN-DeepNN | $0.17 \pm 0.05$ | $\mathbf{0.75} \pm 0.08$ | 2/5 |
| BN-DeepNN* | $0.18 \pm 0.03$ | $0.74 \pm 0.09$ | 2/5 |
| **BN-AFT** | $0.18 \pm 0.04$ | $\mathbf{0.75} \pm 0.06$ | **5/5** |
| **BN-AFT*** | $0.18 \pm 0.04$ | $\mathbf{0.75} \pm 0.06$ | **5/5** |
| CoxPH | $0.41 \pm 0.12$ | $0.63 \pm 0.08$ | 0/5 |
| DeepSurv | $0.21 \pm 0.08$ | $0.73 \pm 0.09$ | 4/5 |
| AFT | $\mathbf{0.16} \pm 0.04$ | $0.72 \pm 0.08$ | **5/5** |
| MTLR | $0.21 \pm 0.08$ | $0.72 \pm 0.10$ | 3/5 |
| **Multiple Visits** | | | |
| BN-CoxPH | $0.22 \pm 0.05$ | $0.72 \pm 0.07$ | 1/5 |
| BN-CoxPH* | $0.22 \pm 0.05$ | $0.73 \pm 0.06$ | 1/5 |
| BN-DeepNN | $0.22 \pm 0.03$ | $0.73 \pm 0.06$ | 2/5 |
| BN-DeepNN* | $0.22 \pm 0.03$ | $0.72 \pm 0.05$ | 2/5 |
| **BN-AFT** | $\mathbf{0.21} \pm 0.02$ | $\mathbf{0.75} \pm 0.06$ | **5/5** |
| BN-AFT* | $\mathbf{0.21} \pm 0.02$ | $\mathbf{0.75} \pm 0.06$ | 4/5 |
| CoxPH | $0.32 \pm 0.03$ | $0.65 \pm 0.07$ | 1/5 |
| DeepSurv | $0.23 \pm 0.03$ | $0.72 \pm 0.06$ | 2/5 |
| AFT | $0.22 \pm 0.06$ | $0.74 \pm 0.08$ | 4/5 |
| MTLR | $\mathbf{0.21} \pm 0.07$ | $0.73 \pm 0.09$ | 2/5 |

(MAE-PO), and D-Calibration. All continuous variables have been normalized by dividing by their maximum values. Therefore, all error values for survival time are normalized values.

**Case 1.** Considering one visit and knowledge-based feature selection resulted in 22 selected features with 57 instances after removing instances with at least one missing value in their features. Table 1 shows the results for different models. Among the models, our Bayesian network approaches (BN-CoxPH, BN-DeepNN, BN-AFT) consistently outperformed traditional baselines across most evaluation metrics. Notably, **BN-AFT** achieved the best overall performance, with the strong concordance (C-index $0.75 \pm 0.06$), and perfect D-Calibration (5/5), indicating both accurate and well-calibrated individual survival distributions. **BN-DeepNN** also performed competitively, particularly in C-index and MAE-PO, suggesting the advantage of neural conditional probability distributions in capturing complex relationships. In contrast, traditional models like CoxPH and AFT showed C-index scores, highlighting the benefit of modeling conditional dependencies between variables. Although DeepSurv and MTLR demonstrated reasonable predictive accuracy, they were generally outperformed by our BN-based methods, particularly in terms of calibration. These results emphasize the strength of Bayesian networks in integrating dependency structure and flexible CPDs for individualized survival prediction.

**Case 2.** Considering multiple visits and knowledge-based features selection leads to 22 final features with 94 instances. Table 1 shows the survival prediction performance when incorporating information from three patient visits. Similar to the one-visit setting, the Bayesian network models continue to demonstrate strong predictive power. **BN-AFT** again achieves the best overall results with the highest D-Calibration score (5/5), while also attaining the highest C-index ($0.75 \pm 0.06$), indicating both accurate and well-calibrated individual survival predictions. While **BN-CoxPH** and **BN-DeepNN** also provide competitive results, particularly in terms of C-index, traditional models such as CoxPH and AFT show notably higher errors and weaker calibration. Although DeepSurv and MTLR maintain reasonable performance in terms of discrimination (C-index), they do not outperform the BN-based models overall. These results reinforce the effectiveness of leveraging temporal information through Bayesian networks for improving individualized survival modeling in longitudinal clinical data. Overall, the three experimental cases highlight the trade-offs between visit

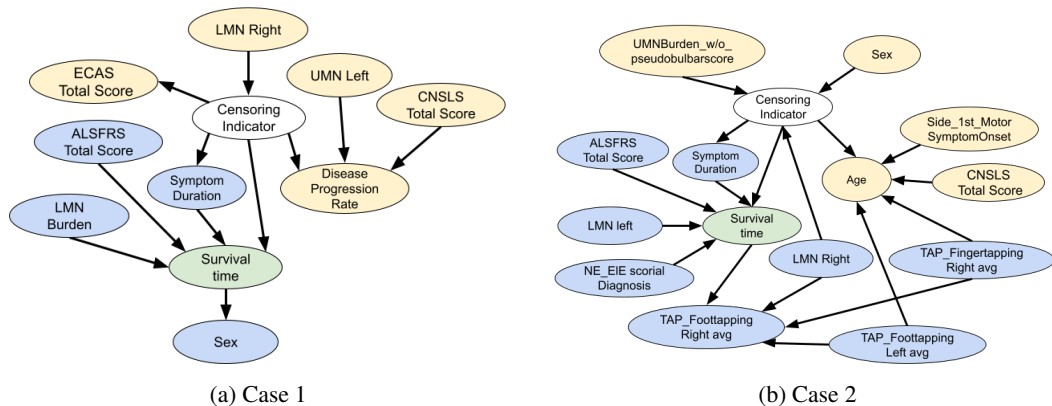

(a) Case 1                                          (b) Case 2

Figure 1: The relevant portion of the Bayesian networks learned in (a) Case 1 and (b) Case 2, illustrating the survival time, censoring indicator, and their corresponding Markov blankets. For ISD estimation, Method 1 uses all blue and yellow nodes, whereas Method 2 only uses blue nodes.

representation and feature selection strategies.

Bayesian network (BN) models, particularly BN-AFT, performed strongly in **Case 1** (single-visit, domain-informed features) despite limited data. Expanding to **Case 2** (three visits as independent instances, same features) improved prediction and calibration, again with BN-AFT leading. In **Case 4** (three visits + data-driven feature selection, larger feature/instance set), deep learning models like DeepSurv achieved the best results, though BN models (especially BN-DeepNN) remained competitive and robust.

Treating multiple visits per patient as independent instances increased sample size but violated independence assumptions, which could bias some models. Nevertheless, BN models showed resilience, and incorporating temporal information improved performance. Future work could explicitly model patient-level dependencies using hierarchical or temporal BNs.

Structural analysis showed that larger sample sizes led to denser Markov blankets (See Fig. 1), but prediction results remained similar. Moreover, **Method 2** achieved comparable survival prediction to **Method 1** while using fewer features (only the survival-time Markov blanket).

## 5 CONCLUSION

There are many tools that use a survival dataset, with censored instances, to learn survival models, for predicting the time to an event — including models that simply return a risk score, or a simple time-based probability (*e.g.*, of dying within the first year), etc. Here, we focus on tools that estimate an Individual Survival Distribution (ISD) $S(t|x)$ for each instance $x$, which gives the probability that $x$ will live at least $t$ (years). There are many such tools here as well – MTLR, AFT, N-MTLR, etc. Unfortunately, most of these tools are opaque, in that it is not clear which covariates are essential to estimating the ISD. This motivated our approach, of modeling the survival prediction as a Bayesian network – as this allows us to identify a subset of covariates that is sufficient to produce an accurate estimate, and also see how these covariates are related to one another! This means we need only collect this reduced subset of features of novel patients, to make accurate forecasts Our empirical study on the ALS task showed that this can work effectively on very challenging data. Moreover, our conversations with practicing neurologists confirm that these features are reasonable here.

This required addressing several challenges – *e.g.*, to deal with the different types of features, to avoid overfitting, etc. While this paper focused on a specific task – predicting the time until death for ALS patients – this is a very general approach, which could potentially be used for any other survival task – perhaps predicting the time until death for other diseases, or time until recovery, or non-medical examples: time until a mechanical part breaks, or until a customer stops shopping at a store. In all of these situations, this technology will produce competitive, accurate results, in a way that allows the user to identify which features are sufficient to produce that survival distribution.

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

Table 2: Survival prediction performance on ALS dataset (mean ± std across folds). Comparison between using only the first visit and using all three visits as independent instances. Feature selection was based on dependence tests.

| Method | MAE-PO ↓ | C-index ↑ | D-Cal. ↑ |
|---|---|---|---|
| **One Visit** | | | |
| BN-CoxPH | $0.36 \pm 0.03$ | $0.70 \pm 0.03$ | 1/5 |
| BN-DeepNN | $0.27 \pm 0.03$ | $0.65 \pm 0.02$ | 3/5 |
| BN-AFT | $0.22 \pm 0.03$ | **0.75** $\pm 0.04$ | **4/5** |
| CoxPH | $0.45 \pm 0.10$ | $0.59 \pm 0.09$ | 0/5 |
| DeepSurv | $0.23 \pm 0.04$ | $0.67 \pm 0.05$ | 3/5 |
| AFT | **0.20** $\pm 0.03$ | $0.74 \pm 0.02$ | 3/5 |
| MTLR | $0.23 \pm 0.04$ | $0.70 \pm 0.03$ | **4/5** |
| **Multiple Visits** | | | |
| BN-CoxPH | $0.31 \pm 0.03$ | $0.74 \pm 0.03$ | 1/5 |
| BN-DeepNN | $0.25 \pm 0.04$ | $0.73 \pm 0.02$ | 2/5 |
| BN-AFT | $0.29 \pm 0.04$ | $0.71 \pm 0.02$ | **3/5** |
| CoxPH | $0.41 \pm 0.02$ | $0.64 \pm 0.05$ | 0/5 |
| **DeepSurv** | **0.22** $\pm 0.04$ | **0.75** $\pm 0.06$ | 2/5 |
| AFT | $0.31 \pm 0.06$ | $0.71 \pm 0.02$ | **3/5** |
| MTLR | **0.22** $\pm 0.05$ | $0.72 \pm 0.06$ | 2/5 |

## A  APPENDIX

### A.1  NUMERICAL RESULTS (CASES 3 AND 4)

**Case 3.** By considering the first visit for each individual and applying feature selection through dependency testing with a significance level of 0.01, this process resulted in the selection of 52 features and 97 instances after removing instances with any missing values. The results in the table show that **BN-AFT** and **AFT** achieve the best overall performance across all three metrics. BN-AFT obtains the highest C-index (0.75) and strong D-calibration (4/5), along with a low MAE-PO (0.22), while AFT achieves the lowest MAE-PO (0.20) and competitive performance in the other metrics. Compared to standard models, the Bayesian network variants – especially BN-AFT – consistently perform better in terms of both calibration and discrimination. **DeepSurv** and **MTLR** also demonstrate balanced performance, though with slightly lower accuracy. In contrast, the traditional **CoxPH** model performs the worst across all metrics. Overall, modeling survival using Bayesian networks, particularly with AFT conditional distributions, improves both predictive accuracy and calibration.

**Case 4.** By considering three visits and applying feature selection through dependency testing, like **Case 3**, 52 features would be selected. After excluding instances with missing values in the selected features, the final dataset for ISD estimation included 150 instances with 52 features, survival time, and censoring indicator. Table 2 reports the survival prediction results across all methods. In this setting, **DeepSurv** achieves the best overall performance, obtaining the highest C-index ($0.75 \pm 0.06$), indicating strong individual survival prediction capabilities. Among the Bayesian network models, **BN-DeepNN** performs best, with competitive MAE and C-index, suggesting the neural CPD's flexibility benefits survival modeling in dynamic settings. However, **BN-AFT** and **BN-CoxPH** show slightly higher errors and poorer calibration compared to DeepSurv and MTLR. Traditional models such as CoxPH continue to lag behind, with significantly higher errors and poor calibration (0/5). Overall, while DeepSurv leads in predictive accuracy in this visit-dependent context, Bayesian network models still offer competitive results, particularly with the integration of flexible CPDs and structural knowledge from the data.

### A.2  EVALUATION METRICS

We evaluate the performance of the survival prediction models using five complementary metrics that assess both the accuracy of the predicted death times and the reliability of the full survival

distributions. All time-point predictions are based on the **median survival time**, defined as the earliest time $\hat{t}_{\mathrm{med}}$ such that the predicted survival function $\hat{S}(t)$ drops below 0.5, i.e., $\hat{S}(\hat{t}_{\mathrm{med}}) \leq 0.5$. See Figure 2.

**1. Concordance Index (C-index)**    The C-index measures the ability of the model to correctly rank individuals by their risk of death. It computes the proportion of all comparable pairs where the individual with the shorter observed survival time is also assigned a shorter predicted median time. Let $\mathcal{P}$ be the set of comparable pairs $(i, j)$ such that $T_i < T_j$ and $\delta_i = 1$ (i.e., $i$ experienced the event before $j$). Then:

$$\text{C-index} = \frac{1}{|\mathcal{P}|} \sum_{(i,j) \in \mathcal{P}} \mathbb{I}[\hat{t}_{\mathrm{med},i} < \hat{t}_{\mathrm{med},j}].$$

**4. MAE on Pseudo-Observations (MAE-PO)**    Qi et al. (2023) To enable performance evaluation over both censored and uncensored instances, we employ pseudo-observations $T_i$ as jackknife-based estimates of marginal survival time, typically derived from the Kaplan–Meier estimator or other consistent survival function estimators. These pseudo-values approximate the expected survival time for each individual, even in the presence of censoring.

The MAE on Pseudo-Observations (MAE-PO) is then defined as:

$$\text{MAE-PO} = \frac{1}{N} \sum_{i=1}^{N} \left| \hat{t}_{\mathrm{med},i} - T_i \right|,$$

where $\hat{t}_{\mathrm{med},i}$ denotes the predicted median survival time for instance $i$, and $T_i$ is the pseudo-observation of survival time.

This metric enables a fair and consistent comparison of survival predictions, particularly median-based predictions, by mitigating the bias introduced by right-censoring.

**D-Calibration.**    Haider et al. (2020) D-calibration evaluates whether the predicted survival distributions are well-calibrated with respect to the observed event times. For each individual $i$ with an uncensored event, let $\hat{S}_i(t)$ denote the predicted survival function. If the model is well-calibrated, then the predicted survival probability at the true event time, $U_i = \hat{S}_i(E_i)$, should be uniformly distributed in $[0, 1]$ across the population.

In practice, if $\hat{S}_i(t)$ is available via Monte Carlo sampling (i.e., $s_i^{(1)}, \ldots, s_i^{(M)} \sim \hat{P}_i(t)$), the survival probability at time $E_i$ is estimated as:

$$U_i = \hat{S}_i(E_i) \approx \frac{1}{M} \sum_{m=1}^{M} \mathbb{I}(s_i^{(m)} > E_i),$$

where $\mathbb{I}(\cdot)$ is the indicator function. These values $\{U_i\}$ are collected for all uncensored individuals.

To assess calibration, we use a histogram-based chi-squared test to check whether $\{U_i\}$ are uniformly distributed. Specifically, the $[0, 1]$ interval is divided into $B$ equal-width bins, and we compute:

$$\chi^2 = \sum_{j=1}^{B} \frac{(O_j - F)^2}{F},$$

where $O_j$ is the observed number of $U_i$ in bin $j$, and $F = N/B$ is the expected count under uniformity. The resulting $\chi^2$ statistic is compared to the chi-squared distribution with $B - 1$ degrees of freedom to obtain a $p$-value.

A high $p$-value (e.g., $> 0.05$) indicates that the predicted distributions are consistent with the observed data, i.e., the model is well-calibrated. A low $p$-value suggests miscalibration—either over- or underestimation of survival probabilities.

A.3    THE USE OF LARGE LANGUAGE MODELS (LLMs)

We used LLMs for polishing the text.

---

**Algorithm 2** ISD Estimation via Joint Markov Blanket Sampling

---

**Input**: BN $\mathcal{B} = (\mathcal{G}^\star, \hat{\boldsymbol{\theta}})$, test covariates $\mathbf{x}_{\text{test}}$, number of samples $M$.
**Output**: Estimated ISD $\hat{S}(t)$.

1: Identify the joint Markov blanket: $\mathcal{M} \leftarrow \text{MB}(E) \cup \text{MB}(\delta) \setminus \{E, \delta\}$.
2: Extract evidence $\mathbf{x}_{\mathcal{M}}$ from $\mathbf{x}_{\text{test}}$.
3: Initialize sample set: $\mathcal{T}_{\text{samples}} \leftarrow []$.
4: **for** $j = 1$ **to** $M$ **do**
5:     Sample $E^{(j)}$ by running Gibbs sampling over the variables $\{E\} \cup \mathcal{M}$, conditioned on the evidence $\mathbf{x}_{\mathcal{M}}$.
6:     Append the sampled time $E^{(j)}$ to $\mathcal{T}_{\text{samples}}$.
7: **end for**
8: Compute ISD: $\hat{S}(t) \approx \frac{1}{M} \sum_{j=1}^{M} \mathbb{I}(E^{(j)} > t)$.
9: **return** $\hat{S}(t)$

---

---

**Algorithm 3** ISD Estimation via Conditional Markov Blanket Sampling

---

**Input**: BN $\mathcal{B} = (\mathcal{G}^\star, \hat{\boldsymbol{\theta}})$, test covariates $\mathbf{x}_{\text{test}}$, number of samples $M$.
**Output**: Estimated conditional ISD $\hat{S}(t)$.

1: Identify the Markov blanket of the time node: $\mathcal{M}_E \leftarrow \text{MB}(E)$.
2: Extract the full evidence set: $\mathbf{e} \leftarrow \mathbf{x}_{\mathcal{M}_E} \cup \{\delta = 1\}$.
3: Initialize sample set: $\mathcal{T}_{\text{samples}} \leftarrow []$.
4: **for** $j = 1$ **to** $M$ **do**
5:     Sample a new time $E^{(j)}$ from its full conditional distribution $P(E \mid \mathbf{e})$ using a targeted Gibbs step.
6:     Append $E^{(j)}$ to $\mathcal{T}_{\text{samples}}$.
7: **end for**
8: Compute ISD: $\hat{S}(t) \approx \frac{1}{M} \sum_{j=1}^{M} \mathbb{I}(E^{(j)} > t)$.
9: **return** $\hat{S}(t)$

---

Table 3: Selected Variables via knowledge domain

| Variable | Type of CPD |
| --- | --- |
| Age | Continuous |
| Sex | Binary |
| Handedness | Categorical |
| YearsEd | Continuous |
| Side_1st_MotorSymptomOnset | Categorical |
| Region_of_Onset | Categorical |
| ALSFRS_TotalScore | Continuous |
| Disease Progress Rate | Continuous |
| TAP_Fingertapping_Right_avg | Continuous |
| TAP_Fingertapping_Left_avg | Continuous |
| TAP_Foottapping_Right_avg | Continuous |
| TAP_Foottapping_Left_avg | Continuous |
| CNSLS_TotalScore | Continuous |
| UMN_Right | Continuous |
| UMN_Left | Continuous |
| UMNBurden_w/o_pseudobulbarscore | Continuous |
| LMN_Right | Continuous |
| LMN_Left | Continuous |
| LMN_Burden | Continuous |
| NE_ElEscorial_Diagnosis | Categorical |
| ECAS_TotalScore | Continuous |
| Symptom Duration | Continuous |
| Censoring indicator | Binary |
| Survival time | Exponential based |

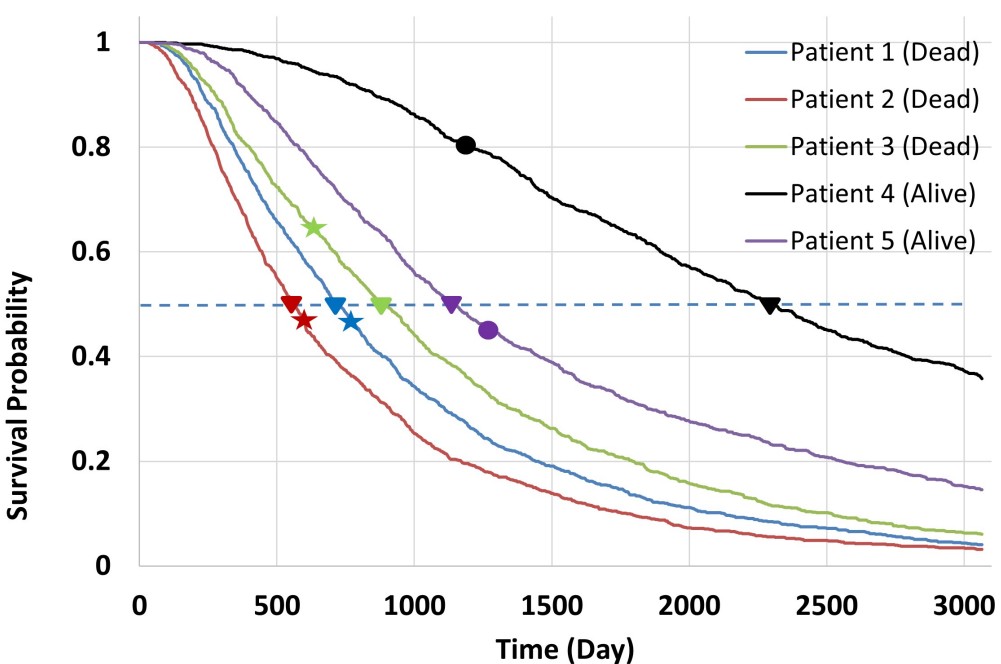

Figure 2: ISDs for five different individuals , all from model BN-AFT in Case 1. Patients 1, 2, and 3 died during the study (and so are uncensored), and patients 4 and 5 are censored, as they were alive at the end of the study. The horizontal dashed line at 0.5 helps to identify the median survival time, and is shown using a ▼. We also use ★ to indicates true survival time, and ● for censoring time. So the median time for patient 2 (red) is 560 days (blue triangle), whereas his true survival time was 606 days (red star); The median for patient 4 (black) was 2307 days (black triangle), and his censored time was 1211 days (black circle), and the median for patient 5 (purple) was 1175 days (purple triangle), but her censored time was 1289 days (purple circle), etc.

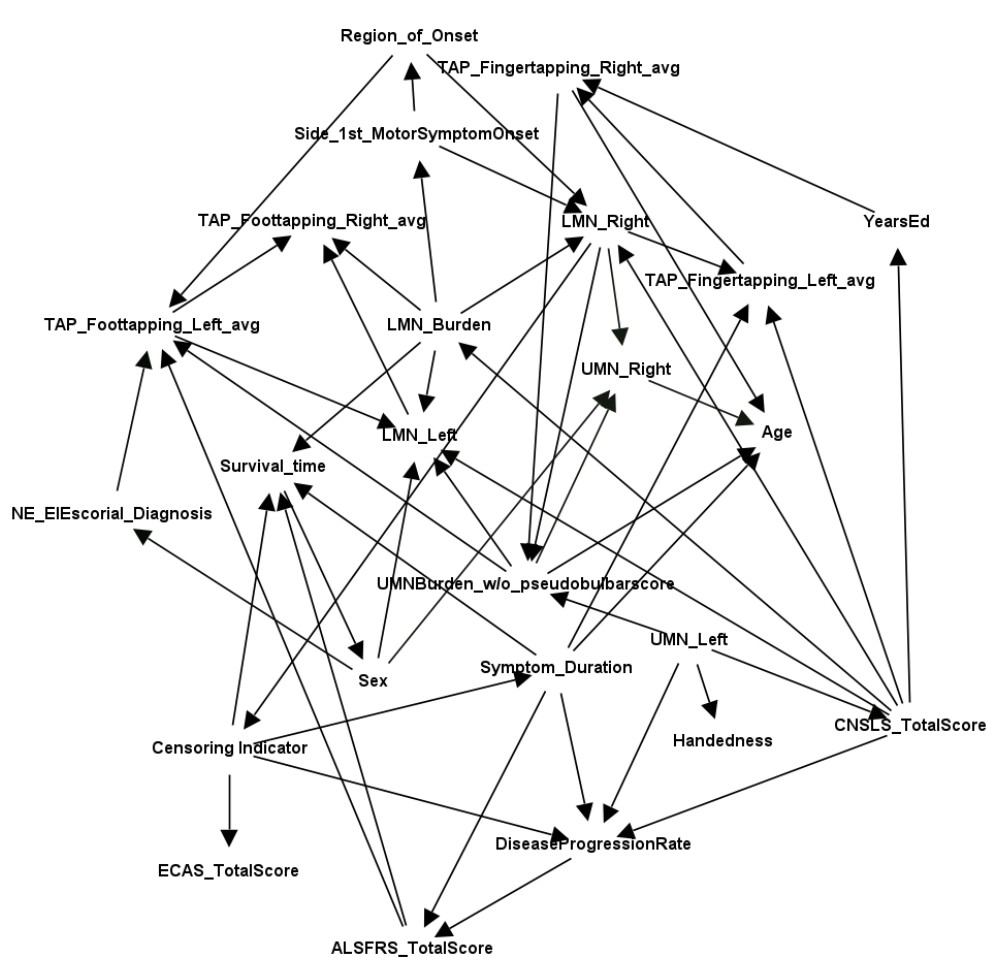

Figure 3: The whole structure for Case 1

