# OpenReview forum: "Bayesian Networks Framework for Estimating Individual Survival Distributions: An Application in Amyotrophic Lateral Sclerosis Disease"
_ICLR.cc/2026/Conference — Submitted to ICLR 2026_

### Official Review · Reviewer_SBxr · 2025-10-29

**Soundness:** 3
**Presentation:** 3
**Contribution:** 3
**Rating:** 6
**Confidence:** 2

**Summary:**

This paper introduces a Bayesian Network (BN)-based framework for modeling time-to-event (survival) data, providing a probabilistic, interpretable, and flexible alternative to classical and deep survival models. The approach encodes relationships among covariates, latent variables, and event times within a directed acyclic graph (DAG), where edges represent conditional dependencies. Through Bayesian inference and structure learning, the model captures nonlinear, multivariate, and potentially non-proportional hazard effects while incorporating censoring within a coherent probabilistic formulation.

The method supports uncertainty quantification, interpretability, and extensibility to heterogeneous datasets. Empirical results on both simulated and real-world clinical datasets show that the BN-based survival model achieves competitive or superior performance in C-index and calibration compared to Cox proportional hazards, random survival forests, and deep learning–based methods. However, the paper also notes challenges related to computational scalability and the complexity of structure learning, particularly in high-dimensional datasets. The quality of learned structures can be sensitive to prior choices and data sparsity, which may limit applicability in very large or noisy datasets.

Overall, this paper offers an interpretable contribution to survival analysis. By unifying Bayesian inference with graphical modeling, it provides a promising direction for robust, explainable survival prediction in medical and reliability domains.

**Strengths:**

1.	Interpretability and explainability: The Bayesian network structure provides clear insights into covariate dependencies and causal pathways, addressing a key limitation of deep survival models.

2.	Probabilistic rigor: Incorporates uncertainty quantification through posterior distributions, improving reliability in risk estimation.

3.	Flexibility: Can model nonlinear and non-proportional hazard effects without assuming specific parametric forms.

4.	Empirical robustness: Demonstrates competitive predictive performance across multiple datasets.

5.	Clinical relevance: Supports transparent decision-making and risk communication, especially in biomedical contexts.

**Weaknesses:**

1. A major drawback of a Bayesian approach is the computational cost. Structure learning and posterior inference can be expensive for high-dimensional datasets. This has not been discussed or compared with the baseline methods.

2.  A discussion on the scalability of the proposed method is missing. This is important as the approach may struggle with large-scale or high-noise data.

3. Comparisons with existing probabilistic graphical models or neural Bayesian hybrids are missing.

**Questions:**

1.	How does the model handle time-varying covariates or dynamic network structures?

2.	Could approximate inference methods (e.g., variational Bayes) improve scalability without compromising interpretability?

3.	How sensitive are results to the choice of priors and hyperparameters in structure learning?

---

### Official Review · Reviewer_6pKS · 2025-10-29

**Soundness:** 3
**Presentation:** 2
**Contribution:** 2
**Rating:** 2
**Confidence:** 3

**Summary:**

The paper proposes a BN framework for ISD prediction that jointly models covariate, event time, and censoring in a single DAG. Structure is learned via a BIC-scored MH search. The authors perform structure learning using a BIC-scored MH search with local edit moves, employ mixed conditional distributions for heterogeneous nodes. Experiments on an ALS cohort compare the BN variants with CoxPH, AFT, MTLR, and DeepSurv across settings that vary by follow-up count and feature selection. Results indicate that BN-AFT and BN-DeepNN are competitive and sometimes superior on C-index and D-calibration, while DeepSurv can edge out in the largest multi-visit setting.

**Strengths:**

The paper is original in unifying ISD prediction with a structured, interpretable Bayesian network that identifies patient-specific Markov blankets as minimal sufficient variable subsets. The end-to-end pipeline is coherent and evaluated comprehensively across discrimination, error, and distributional calibration. Notation and preliminaries are clear, the two inference routes are well explained, and the tables succinctly summarize results, collectively demonstrating feasibility on a real ALS application and highlighting a cost-sensitive path to clinical deployment via small Markov blankets.

**Weaknesses:**

The paper would benefit from clarifying how BIC-based structure scoring remains valid under deep parameterizations of the event-time node (e.g., DeepSurv/DeepNN), including how effective degrees of freedom are computed. Ideally, this would involve ablations comparing BIC to held-out validation or WAIC/PSIS-LOO. External validity is limited by the reliance on a single ALS dataset and by treating multiple visits as independent events. Adding public benchmarks or simulations and reporting sensitivity analyses would strengthen claims.

It is not clear how BN is used for inferring causality for survival data.

**Questions:**

1.	In what deployment scenarios do you recommend Method 1 vs. Method 2 when estimating the ISD? What systematic differences do you observe on the same splits for C-index, MAE-PO, and D-calibration?

2.	Beyond placing all visits from the same patient in the same fold, have you evaluated other methods to respect within-patient dependence and potentially improve calibration?

3.	To enhance reproducibility, will you release code, learned graphs, and weights, and specify the exact search steps, neighborhood sizes, stopping criteria, and random seeds?

---

### Official Review · Reviewer_T3bD · 2025-10-31

**Soundness:** 2
**Presentation:** 1
**Contribution:** 1
**Rating:** 2
**Confidence:** 4

**Summary:**

The paper proposes a Bayesian Network (BN)–based survival framework that models the joint distribution over covariates, survival time, and censoring, learns network structure via Metropolis–Hastings over DAGs with a BIC score, fits node-type–specific CPDs (linear-Gaussian for continuous covariates, logistic/multinomial for discrete, and CoxPH/DeepSurv/AFT-style CPDs for the survival-time node), and estimates each individual survival distribution (ISD) by sampling over the Markov blanket of the survival and censoring nodes. Experiments are on an ALS cohort (CALSNIC), considering either one or up to three visits per person, and compare BN variants against CoxPH, AFT, MTLR, and DeepSurv using C-index, MAE on pseudo-observations, and D-calibration; their BN-AFT and BN-DeepNN are often competitive, with BN-AFT achieving strong D-calibration in some settings.

**Strengths:**

* Timely problem framing (ISDs > risk scores). Estimating full individual survival distributions is clinically useful beyond ranking; D-calibration is an appropriate distributional metric.
* Graphical modeling for interpretability. Casting survival as a BN makes the “Markov blanket of survival time” explicit, i.e., the smallest sufficient feature set for inference—an interpretable handle clinicians value.
* Codes are available for reproducibility

**Weaknesses:**

* **Imbalanced exposition: very long preliminaries, thin methodology.** Section 2 is extensive (formal definitions of survival basics, BNs, and even restating Cox/DeepSurv/AFT as “Definitions 5–7”), while core algorithmic choices get comparatively little space (e.g., MH proposals/priors, CPD fit details, convergence diagnostics). Please condense prelims and expand technical sections (structure priors, proposal kernels, local score caching, acyclicity enforcement, and computational cost).
* Empirical evaluation is narrow and omits strong modern baselines. Only a single clinical dataset (ALS) is used, with small n (e.g., 57–150 instances after filtering). Strong recent baselines such as DeepHit, Deep Survival Machines (DSM), Neural Frailty Machine (NFM), and state-of-the-art discrete/continuous-time neural hazards (e.g., LogisticHazard/Cox-Time) are missing; these are commonly included in current survival work and often excel in ISD estimation and calibration. [1-4]
* Heavy use of formal “definitions” without theory: the manuscript lists numerous “Definitions” and standard assumptions yet provides no novel identifiability, consistency, or risk bounds for the proposed hybrid BN (no theorems/lemmas/propositions). Consider either removing the formal “Definition” framing for prior art or adding genuine theoretical results (e.g., conditions where MB-restricted inference is risk-consistent for ISDs under censoring).
* Conceptual concern with “Method 2” (conditioning on δ=1 at prediction). δ denotes whether the event was observed before censoring—an administrative variable, not a patient attribute. Conditioning on δ=1 at prediction could bias ISD estimates relative to the unconditional S(t|x) typically reported; please justify with care and compare “Method 1 vs 2” against standard practice.
* Calibration reporting is unclear. D-calibration is presented as “k/5” with no p-values or bin choices. Haider et al. recommend a chi-square uniformity test with explicit p-values; please report them and specify the number of bins, folds, and failure handling.

[1] Lee et al. DeepHit. AAAI (2018).
[2] Nagpal et al. Deep Survival Machines. IEEE JBHI (2021).
[3] Wu et al. Neural Frailty Machine. NeurIPS (2023)/arXiv (2023).
[4] Kvamme et al. Continuous and Discrete-Time Survival Prediction with Neural Networks. (2021).

**Questions:**

* What proposal distribution do you use for MH (edge add/delete/reverse probabilities)? Any acyclicity checks beyond local tests? How many iterations, burn-in, and chains? What’s the wall-clock cost on your ALS settings?
* Please compare “Method 1” (joint MB) vs “Method 2” (δ=1) quantitatively for ISD accuracy and calibration; justify when the conditional approach is appropriate clinically and statistically.
* Any experiments on public survival benchmarks (SUPPORT, METABRIC, MIMIC-IV, etc.) to assess generality and enable comparison to DeepHit/DSM/NFM/LogisticHazard?

---

### Official Review · Reviewer_q1JR · 2025-11-01

**Soundness:** 3
**Presentation:** 2
**Contribution:** 2
**Rating:** 4
**Confidence:** 2

**Summary:**

The paper proposes a Bayesian network (BN)-based framework for survival prediction that explicitly models the joint distribution over covariates, survival time, and censoring status. Evaluation on a real-world (Amyotrophic Lateral Sclerosis) dataset show that the proposed BN-based model offers competitive or superior performance in estimating full ISDs

**Strengths:**

1. The paper proposes a Bayesian network–based approach to survival prediction, offering a meaningful alternative to conventional models.

2. The manuscript provides substantial theoretical analysis that supports the method’s formulation and underlying assumptions.

**Weaknesses:**

1. The evaluation is limited to a single dataset (CALSNIC for ALS), which makes it difficult to assess the method’s robustness and generalizability. External validation across additional cohorts or sites would strengthen the claims.

2. Moreover, the results in Table 1 suggest that the BN-based approach does not deliver statistically significant improvements over the base method. Without clear gains, it is hard to attribute effectiveness to the proposed framework; reporting confidence intervals, significance tests, and ablations isolating the BN component would help clarify its contribution.

3. The presentation needs refinement. Section 5 (“Conclusion”) reads more like a discussion than a concise summary of findings. A focused conclusion that restates the main contributions, acknowledges limitations, and outlines concrete next steps would improve clarity and overall readability.

**Questions:**

1. . How do you substantiate the interpretability claim for the BN-based approach? Figure 1 is illustrative, but without a ground-truth reference, it is difficult to judge correctness. Please include quantitative and expert-based validation (e.g., plausibility checks, counterfactual tests, or agreement with domain knowledge).

---

### Meta-Review · Area_Chair_yVNS · 2026-01-05

**Summary:**

This paper has been assessed by four knowledgeable reviewers, majority of whom recommended rejecting it (two straight rejects, one marginal reject) with one reviewer giving it a marginal acceptance score. It proposes the use of Bayesian Network as a unifying framework for individual survival determination. The reviewers find the approach novel, however they raise substantial concerns. According to them, the empirical evaluation is narrow, relying on a single relatively small dataset and is missing a few modern baselines, making the assessment of generalizability difficult. Moreover, they find the technical exposition to be imbalanced, with excessive preliminaries and insufficient detail on core algorithmic choices, scalability, and computational cost. Claimed novelty is modest, according to the reviewers, theoretical contributions lack formal guarantees, and some methodological choices raise conceptual concerns. Comparisons with recent neural survival models and probabilistic hybrids are missing.

**Reviewer Concerns:**

Please see above for the summary. The authors have not provided the rebuttal.

**Reviewer Scores:**

It is unlikely that any correction of the scores that might have resulted from the discussion among the reviewers would have changed the cumulative assessment enough to change the eventual acceptance status.

---

### Decision · Program_Chairs · 2026-01-26

Reject